# Opioid-sparing anesthesia versus opioid-free anesthesia for postoperative recovery quality in breast cancer surgery patients: A systematic review and Bayesian network meta-analysis

JingWang Liu, JiaXin Liu, MingJie Wang, XiuLi Wang*

Department of Anesthesiology, The Third Hospital of Hebei Medical University, Shijiazhuang, Hebei, China

* wangxl311@126.com

## Abstract

### Background

Breast cancer is the most common malignancy among women, and patients require rapid transition to adjuvant therapy post-surgery. Opioid-based anesthesia (OBA) is widely used but carries risks such as postoperative nausea and vomiting (PONV), immunosuppression, and hyperalgesia, which may delay recovery. Opioid-sparing anesthesia (OSA) and opioid-free anesthesia (OFA) may reduce these risks, but their effects on postoperative quality of recovery (QoR) are unclear. This study compares the effects of these three anesthetic strategies on early postoperative QoR in breast cancer surgery using a Bayesian network meta-analysis.

### Methods

Following PRISMA guidelines (PROSPERO ID: CRD420251065588), a systematic search was conducted in PubMed, Cochrane Library, EMBASE, and Web of Science from inception to June 1, 2025. Included were randomized controlled trials (RCTs) comparing OSA, OFA, and OBA in adult breast cancer surgery that reported QoR scores. Risk of bias and evidence quality were assessed using the Cochrane tool and GRADE system. Bayesian random-effects analysis was performed with the R package *gemtc*. Continuous data were reported as mean differences, and categorical data as odds ratios.

### Results

Seventeen RCTs with 1,254 patients were included. Bayesian network meta-analysis showed that OSA significantly outperformed OBA in 24-hour postoperative QoR (d = 0.050, 95% CrI: 0.038–0.062; SUCRA = 85.3%), and OFA was also superior to OBA (d = 0.044, 95% CrI: 0.020–0.068; SUCRA = 64.7%). No significant difference was found between OSA and OFA (d = −0.006, 95% CrI: −0.029–0.018). Secondary

**Data availability statement:** All Code and raw data files are available from the Github database https://github.com/whenthereis/away.git.

**Funding:** The author(s) received no specific funding for this work.

**Competing interests:** The authors have declared that no competing interests exist.

outcomes SUCRA showed that OFA was most effective in controlling postoperative nausea and vomiting (99.4%) and pain management (81.4%), while OSA excelled in emotional well-being (96.2%) and physical comfort (76.6%). For physical independence, OFA (85.1%) outperformed OSA (63.5%), with no differences in psychological support. Intraoperative opioid reduction showed an inverted U-shaped relationship with QoR improvement (p = 0.0004).

## Conclusion

OSA is the optimal strategy for enhancing overall quality of recovery within 24 hours after breast cancer surgery. Although OFA excels in PONV reduction and pain control, OSA offers more balanced benefits across multiple QoR dimensions. An individualized anesthetic approach is recommended, aiming for opioid minimization rather than complete elimination.

## Introduction

Breast cancer, the most commonly diagnosed malignancy among women worldwide, continues to pose a growing disease burden and has become a serious public health challenge. According to the latest statistics, the annual incidence of breast cancer has surpassed that of lung cancer among female cancer patients, securing its position as the most prevalent cancer and earning the title of the "number one killer of women's health" [1]. Encouragingly, advancements in surgical techniques, including modified radical mastectomy and breast-conserving surgery, have significantly improved tumor eradication rates. However, the physical and immunological toll of prolonged adjuvant chemotherapy after surgery remains a major challenge for patients. Studies have confirmed that the quality of early postoperative recovery is closely associated with long-term rehabilitation outcomes within 1–3 months [2,3], making the optimization of early quality of recovery (QoR) crucial for ensuring continuity of care. In this clinical context, enhancing perioperative management—particularly the precise selection of anesthetic strategies—has become a key clinical issue in bridging surgical treatment with subsequent anti-tumor therapy.

Currently, opioid-based anesthesia remains the mainstay for breast cancer surgeries. These agents provide potent analgesia via μ- and κ-opioid receptor agonism. However, growing concerns have emerged regarding the limitations of opioids. Besides common perioperative discomforts such as nausea, vomiting, and pruritus, opioids may also contribute to central sensitization, increasing the risk of chronic postoperative pain. Furthermore, their immunosuppressive effects may promote tumor cell proliferation, posing a potential threat to the long-term prognosis of breast cancer patients [4,5]. These drawbacks have driven clinical exploration into superior analgesic alternatives. Consequently, opioid-sparing anesthesia (OSA) and opioid-free anesthesia (OFA) have gained increasing attention. Utilizing multimodal analgesia strategies combined with ultrasound-guided regional nerve blocks, both OSA and OFA effectively manage perioperative pain while significantly reducing opioid-related adverse effects. Given the

superficial surgical field and limited tissue trauma characteristic of breast procedures—with postoperative pain primarily arising from incisional and axillary lymph node dissection–related nerve traction, often presenting as mild to moderate in intensity [6]—the reduction or elimination of opioid use is particularly feasible. The close alignment between surgical features and anesthetic strategies allows OSA and OFA to meet analgesic demands while avoiding opioid-related complications. Existing studies have demonstrated that these anesthetic approaches promote faster recovery of gastrointestinal function, shorten hospital stays, and preserve immune function [7,8], aligning well with the principles of enhanced recovery after surgery.

Nevertheless, most research on opioid alternatives has focused on objective recovery indicators such as postoperative nausea, vomiting, and pain control, with relatively limited attention paid to patients' subjective recovery experiences [9,10]. The postoperative QoR score, a crucial tool for assessing recovery quality, encompasses five domains: physical comfort, emotional state, pain management, ability to perform daily activities, and social support. This metric provides a comprehensive quantification of patients' subjective recovery experiences, effectively addressing the limitations of traditional objective indicators in evaluating individual experiences [11]. Notably, although OFA and OSA demonstrate advantages in minimizing adverse effects, whether these improvements in objective indicators can translate into enhanced subjective recovery remains controversial. Despite several randomized controlled trials (RCTs) examining the impact of these two anesthetic strategies on postoperative recovery [12,13], a comprehensive systematic meta-analysis is still lacking. Furthermore, comparative evaluations of OFA and OSA in terms of subjective recovery quality based on evidence-based medicine are insufficient, contributing to uncertainty in clinical anesthetic decision-making. In light of this, the present study employs a Bayesian network meta-analysis to systematically synthesize existing clinical evidence, assess the effects of OFA and OSA on postoperative QoR scores in breast cancer surgery patients, and explore the applicability and relative advantages of these two approaches, thereby providing evidence-based support for precise anesthetic decision-making in clinical practice.

## Methods

This study was conducted following the PRISMA reporting guidelines for Bayesian network meta-analyses and systematic reviews. All data included in this network meta-analysis (NMA) were obtained from publicly available sources, and therefore, ethical approval was not required. The protocol for this NMA was registered with PROSPERO on June 2, 2025, under the registration number: CRD420251065588.

### Inclusion and exclusion criteria

This study included RCTs that met the following criteria:1. The study population consisted of female patients undergoing breast cancer surgery, aged 18 years or older. 2. The intervention involved at least one of the following: OFA or OSA. OFA was defined as an anesthetic protocol that completely avoids the use of opioids intraoperatively, while OSA was defined as an anesthetic strategy that reduces perioperative opioid consumption through specific drugs or techniques. 3. The control group received opioid-based anesthesia (OBA), defined as an anesthetic protocol in which opioids were the primary agents for intraoperative analgesia. 4. The primary outcome was postoperative QoR score within 24 hours. Secondary outcomes included subscales within the QoR score, incidence of postoperative nausea and vomiting (PONV) within 24 hours, and the correlation between the percentage reduction in opioid use and postoperative recovery quality. Exclusion criteria were as follows: 1. Studies published only in the form of letters or abstracts. 2. Studies for which the full text was unavailable. 3. Duplicate publications. 4. Studies from which QoR data could not be extracted.

### Search strategy

This study searched the following databases: PubMed, Cochrane Library, EMBASE, and Web of Science, covering the period from the inception of the databases until June 1, 2025. A comprehensive search was conducted using both subject headings and free text terms. The detailed search strategy can be found in Supplementary Material (S1 Text).

Two independent researchers performed the literature search and assessed the eligibility of the studies identified. For studies that were potentially eligible based on the preliminary screening, the full text of the articles was obtained. Additionally, for records where eligibility could not be clearly determined during the title and abstract screening phase, further data acquisition was carried out. In cases where there was disagreement between the reviewers during the screening process, a third researcher was involved in the final decision to ensure consistency and fairness in the selection process.

## Data extraction

Data were independently extracted by two researchers, with any discrepancies resolved through discussion or by a third researcher acting as arbitrator. Extracted information included: author, year of publication, country, sample size, baseline characteristics (age, BMI), type of surgery, anesthesia protocols, postoperative quality of recovery scores within 24 hours, occurrence of PONV within 24 hours and Intraoperative opioid consumption.For continuous data with a normal distribution, the mean and standard deviation were directly recorded. For non-normally distributed data, the median along with interquartile range or full range was extracted. For dichotomous outcomes, the number of events and total sample size were documented. In cases of missing data, attempts were first made to contact the corresponding authors via email; if unsuccessful, missing values were imputed using the CARE formula or multiple imputation methods.All data were cross-checked by both researchers and then converted into a paired format compatible with the 'gemtc' package to ensure the feasibility of the network meta-analysis.

## Risk of bias assessment

Two researchers independently assessed the quality of the included RCTs using the Cochrane Risk of Bias Tool (version 1). Potential sources of bias were evaluated across seven key domains:1. Random sequence generation;2. Allocation concealment;3.Blinding of participants and personnel;4. Blinding of outcome assessors;5. Incomplete outcome data;6. Selective reporting bias;7. Other sources of bias.Any disagreements were resolved through discussion with a third researcher. If any single domain was judged to be at high risk of bias, the study was considered to be at high overall risk of bias. Publication bias was assessed using funnel plots for the outcome measures in conjunction with Egger's test.

## Assessment of quality of evidence

The same procedure was applied for evaluating the quality of evidence. The GRADE system was used to assess the certainty of the evidence for both primary and secondary outcomes. This system evaluates the certainty of evidence across five domains: risk of bias, inconsistency, indirectness, imprecision, and publication bias, providing a comprehensive appraisal for all outcome comparisons.

## Statistical analysis

This study employed a Bayesian random-effects network meta-analysis framework, evaluated using the 'gemtc' package in R version 4.5.0. The primary outcome was the total score of the QoR scale at 24 hours post-surgery. Secondary outcomes included the five subscales of the QoR at 24 hours, the incidence of PONV within 24 hours, and the correlation between the percentage reduction in intraoperative opioid consumption and the postoperative QoR score at 24 hours.

All QoR score data were modeled based on the mean and standard deviation from the original QoR-15 or QoR-40 scales. For different versions of the scales, the raw data were directly entered into the network meta-analysis according to the 'gemtc' package requirements, followed by linear transformation to standardize the data to a 0–1 range. For skewed data reported as medians, interquartile ranges, or extreme values, the MLN method was used to estimate the mean and standard deviation. This method does not require pre-specifying the candidate distribution type and significantly improves the estimation accuracy of means and standard deviations in non-normally distributed data through Box-Cox transformation combined with maximum likelihood estimation [14]. The implementation was completed using the online tool at https://

smcgrath.shinyapps.io/estmeansd/. If studies reported means ± standard deviations, these were directly used for model input. In multi-arm studies, groups that met the three-group definition were combined based on clinical homogeneity, and Cochran weighting was used to calculate the combined mean and standard deviation.

The Bayesian model assumed default distributions for treatment effects and heterogeneity standard deviations. Consistency testing was performed using node-splitting analysis. If there was no statistically significant difference between direct and indirect comparisons, consistency was considered good, and the model was used for analysis. If inconsistency was found, the sources of inconsistency were analyzed, and after excluding influencing factors, an adjusted model was used. If the cause of inconsistency could not be determined, the consistency model was used. Markov Chain Monte Carlo (MCMC) simulations were run with 4 independent chains, a burn-in period of 20,000 iterations, and 100,000 official samples with a thinning interval of 20 to reduce autocorrelation. Convergence was assessed using the Gelman-Rubin statistic, trace plots, and effective sample size. PONV was analyzed as a binary variable (occurrence/no occurrence), with effect sizes presented as odds ratios (OR) and their 95% credible intervals (CrI). All outcomes were incorporated into network models, with the results visualized using forest plots, cumulative ranking curves, and SUCRA plots.The percentage reduction in intraoperative opioid consumption was treated as a continuous covariate in the network meta-regression, to assess its linear or nonlinear relationship with QoR improvement. The formula for calculating the reduction percentage in opioid consumption was:

$$\text{Reduction percentage} = [(\text{control group consumption} - \text{intervention group consumption})/ \\ \text{control group consumption}] \times 100\%$$

with all opioid consumption data converted to morphine-equivalent doses, and the total intraoperative consumption was used for calculation.

Heterogeneity was assessed using the heterogeneity standard deviation (sd.d) and the $I^2$ statistic. Sensitivity analyses included the exclusion of high-risk bias studies, multi-arm studies, robustness tests of effect sizes, validation of the fixed-effects model, and prior distribution checks. The analysis code and data conversion scripts are publicly available on the GitHub repository https://github.com/whenthereis/away.git.

## Results

### Search results

A total of 1,749 articles were initially identified. After removing duplicates, 471 records remained. Based on titles and abstracts, 95 potentially relevant studies were selected for full-text review. Two independent reviewers assessed the full texts against the predefined inclusion and exclusion criteria, and finally, 17 RCTs were included in the analysis. Among these, 8 studies used the QoR-15 scale as the primary outcome, and 9 studies used the QoR-40 scale. The detailed literature selection process is illustrated in Fig 1 (Fig 1. PRISMA flow diagram). For the specific reasons for literature exclusion, please refer to the supplementary materials (S2 Text).

### Characteristics of included studies

The characteristics of the included studies are summarized in Table 1. A total of 17 RCTs involving 1,254 patients with breast cancer were included. The mean age and BMI were 49.6 ± 10.9 years and 24.2 ± 3.4 kg/m² in the OFA group, 50.2 ± 9.5 years and 24.3 ± 4.0 kg/m² in the OSA group, and 51.8 ± 9.6 years and 24.5 ± 4.6 kg/m² in the OBA group. The studies were published between 2016 and 2025, with 76% published in the past five years. Surgical procedures included modified radical mastectomy, total or partial mastectomy combined with sentinel lymph node biopsy or axillary lymph node dissection.

Thirteen studies compared OSA with OBA, two studies compared OFA with OBA, and two studies compared OFA with OSA. Among the OSA studies, 10 utilized nerve blocks, while 3 used non-opioid analgesics, including lidocaine in two

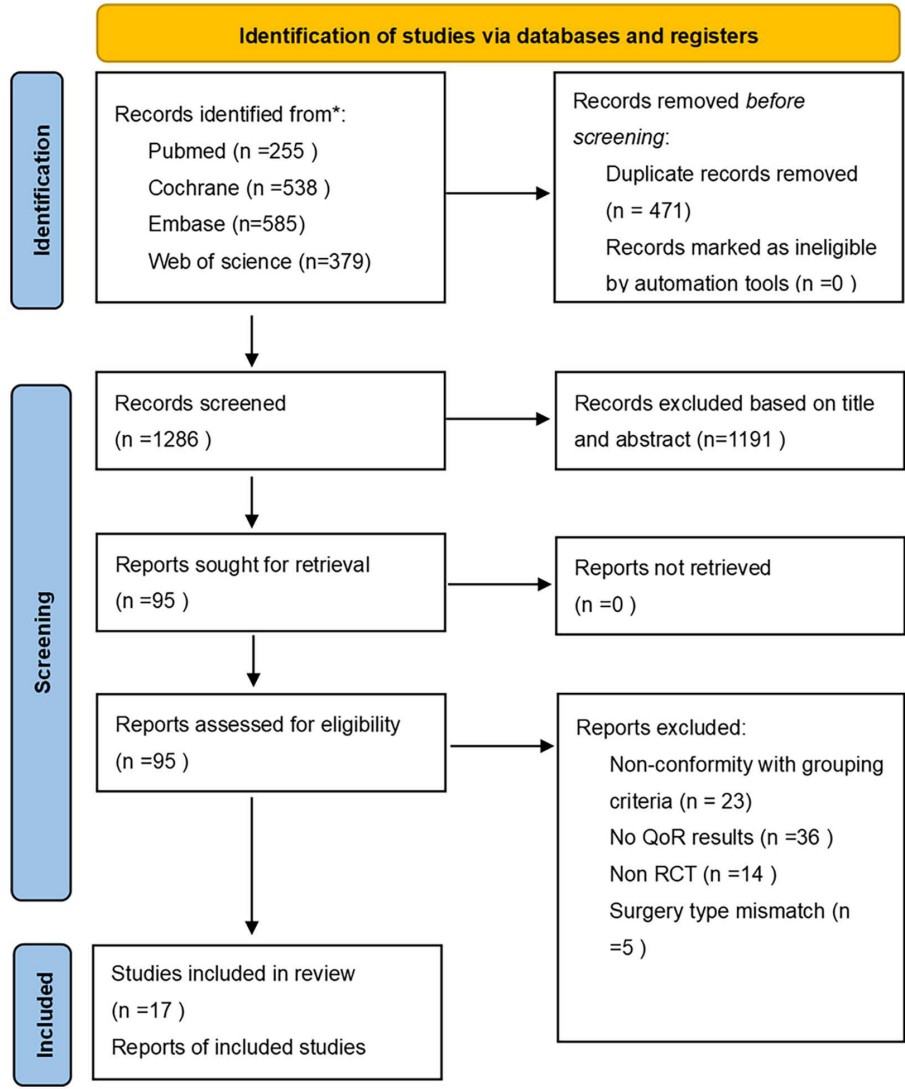

**Fig 1. PRISMA flow diagram.**

studies and ketamine in one. In the four OFA studies, the main pharmacologic agents included dexmedetomidine, ketamine, and lidocaine; of these, two studies applied nerve blocks and two did not.

## Risk of bias assessment

The risk of bias for individual studies and the overall assessment are summarized in Figs 2 and 3. Nine studies were judged to have a low risk of bias. One study was rated as high risk due to the absence of information on blinding of postoperative outcome assessors and lack of description regarding patient blinding.Among the remaining seven studies, one did not report the method for random sequence generation; in three studies, blinding of personnel was not feasible due to the nature of the intervention; two studies did not provide sufficient details on allocation concealment; and one study modified its primary outcome after trial initiation without updating the trial registration and had a small sample size. All of these were judged to have an unclear risk of bias.

**Table 1. Characteristics of included studies (n = 17).**

| Study | Country | Age | BMI | Surgery | Sample | Same strategy | Experimental group strategy | Control group strategy | Postoperative medication | Recovery scale | Results | PONV24 |
|---|---|---|---|---|---|---|---|---|---|---|---|---|
| Başak Altıparmak 2020 [15] | Turkey | OSA:53.8±11.2<br>OBA:52.0±11.5 | OSA:28±2.8<br>OBA:27±3.4 | Modified Radical Mastectomy | OSA:28<br>OBA:28 | Desflurane<br>N₂O<br>Fentanyl<br>Propofol | Ultrasound-guided rhomboid-intercostal nerve block:30 mL of 0.25% bupivacaine | Without nerve block | Morphine PCIA with a 1 mg bolus dose, Dexketoprofen 50 mg IV if VAS≥4 | QoR-40 at POD 24 h | OSA:164.8±3.9<br>OBA:153.5±5.2 | OSA:2 in 27<br>OBA:6 in 27 |
| F. W. Abdallah 2021 [16] | Canada | OSA:58.4±11.8<br>OBA:57.3±12.7 | OSA:25.7±5.5<br>OBA:25.4±3.0 | Mastectomy or BCS combined with SLNB | OSA:20<br>OBA:20 | Desflurane<br>Propofol<br>Fentanyl<br>Hydromorphone | Ultrasound-guided D-SAPB: 20 ml of 0.5% ropivacaine with epinephrine (1:400,000) | Subcutaneous injection of 1 ml normal saline | When VAS≥4, IV fentanyl 12.5–25 µg, repeat every 5 min, max 100 µg. If insufficient, switch to IV hydromorphone 0.2–0.4 mg, max 2 mg. | QoR-15 at POD 24 h | OSA:122±17<br>OBA:121±21 | OSA:6 in 20<br>OBA:9 in 20 |
| Myoung Hwa Kim 2017 [17] | South Korea | OSA1:48.7±6.4<br>OSA2:48.1±7.5<br>OBA:49.0±6.9 | OSA1:22.7±1.6<br>OSA2:22.4±2.1<br>OBA:22.3±3.1 | Mastectomy or BCS combined with SLNB or ALND | OSA1:39<br>OSA2:38<br>OBA:39 | Desflurane<br>Remifentanil<br>Propofol | OSA1: Lidocaine 2 mg/kg IV bolus within 15 min post-induction, then 2 mg·kg⁻¹·h⁻¹ until end of surgery. OSA2: MgSO₄ 20 mg/kg IV over 15 min post-induction, then 20 mg/kg/h until end of surgery. | IV infusion of equal volume normal saline | Not mentioned | QoR-40 at POD 24 h | OSA1:179.3±6.8<br>OSA2:176.6±5.6<br>OBA:173.9±8.4 | Not recorded |
| Aravindhan Krishnasamy Yuvaraj 2023 [18] | India | OFA:48.2±9.8<br>OSA:44.2±6.9 | OFA:23.2±2.3<br>OSA:24.1±3.1 | Mastectomy or BCS combined with SLNB or ALND | OFA:30<br>OSA:30 | Sevoflurane<br>D-SAPB | Pre-induction: dexmedetomidine 1 µg/kg, ketamine 0.3 mg/kg, lignocaine 1.5 mg/kg; intraoperative dexmedetomidine 0.4 µg/kg/h infusion. | Pre-induction: fentanyl 2 µg/kg; intraoperative: fentanyl 0.5 µg/kg/h.. | Postoperatively within 24h: OFA group received dexmedetomidine 0.4 µg/kg/h; OSA group received fentanyl 0.5 µg/kg/h, both via intravenous infusion. | QoR-40 at POD 24 h | OFA:180(176–184)<br>OSA:182(178–186) | Not recorded |

*(Continued)*

Table 1. (Continued)

| Study | Country | Age | BMI | Surgery | Sample | Same strategy | Experimental group strategy | Control group strategy | Postoperative medication | Recovery scale | Results | PONV24 |
|---|---|---|---|---|---|---|---|---|---|---|---|---|
| Qingfen Zhang 2023 [19] | China | OFA:52.6±11.6 OBA:49.2±10.1 | OFA:24.5±3.3 OBA:23.3±3.6 | Mastectomy or BCS combined with SLNB or ALND | OFA:40 OBA:40 | Propofol Sevoflurane | Induction with T2/lidocaine (1.5mg/kg), intraoperative dexmedetomidine infusion (0.4 µg/kg/h), combined with T2/T4 TPVB (40ml of 0.4% ropivacaine). | Induction: sufentanil 0.2–0.3 µg/kg IV; intraoperative: 0.05–0.1 µg/kg as needed. | Flurbiprofen 50mg IV every 12h; breakthrough pain: morphine 1–2mg IV. | QoR-15 at POD 24 h | OFA:140.3±5.2 OBA:132.0±12.0 | OFA:1 in 40 OBA:13 in 40 |
| Fudong Rao 2021 [20] | China | OSA:53.6±6.2 OBA:53.4±6.6 | OSA:22.5±3.7 OBA:22.9±4.1 | Modified Radical Mastectomy | OSA:34 OBA:34 | Remifentanil Sevoflurane Propofol Sufentanil | Ultrasound-guided TPVB: 20mL 0.5% ropivacaine. | Sham nerve block | Parecoxib 40mg IV q12h; PCIA with morphine: demand dose of 2 mg | QoR-40 at POD 24 h | OSA:173(170–177) OBA:161(160–164) | OSA:2 in 34 OBA:8 in 34 |
| Qingfeng Wei 2022 [21] | China | OSA:50.0±9.5 OBA:50.7±8.5 | OSA:24.5±2.3 OBA:24.4±2.3 | Mastectomy or BCS combined with SLNB or ALND | OSA:30 OBA:30 | Propofol Sufentanil Remifentanil | Pre-induction: 2% lidocaine 1.5mg/kg IV over 10min; post-induction: 2.0mg/kg/h IV until surgery end. | IV infusion of equal volume normal saline. | If postoperative VAS>4, IV flurbiprofen axetil 50mg; if VAS remains >4, IV fentanyl 0.1mg. | QoR-15 at POD 24 h | OSA: 128.50±20.25 OBA: 117.50±19.50 | OSA:2 in 30 OBA:5 in 30 |
| Zhang Junxia 2023 [22] | China | OSA:52.6±10.9 OBA:54.9±9.9 | OSA:23.3 (22.2–24.2) OBA:23.7 (22.5–25.9) | Modified Radical Mastectomy | OSA:40 OBA:40 | Propofol Remifentanil | Induction: 0.5mg/kg S-ketamine replacing sufentanil; maintenance: combined continuous infusion of remifentanil and S-ketamine until 30min before surgery end. | Induction with sufentanil 0.3 µg/kg; maintenance with remifentanil 0.2–0.6 µg/kg/min. | If VAS score >3, give diclofenac 75mg IM. | QoR-15 at POD 24 h | OSA:124.0 (119.5–128.0) OBA:119.0 (114.0–123.5) | Not recorded |
| Sihui Zhu 2024 [23] | China | OSA:53(49–60) OBA:51 (45.3–57.6) | OSA:23.24±2.8 OBA:23.9±3.2 | Mastectomy combined with SLNB or ALND | OSA:33 OBA:32 | Propofol Sufentanil Remifentanil Dexmedetomidine | Ultrasound-guided MINB: 30ml of 0.33% ropivacaine. | Sham nerve block | Flurbiprofen axetil 50mg IV every 12 hours; an additional 50mg of flurbiprofen axetil was administered IV if VAS score >3. | QoR-15 at POD 24 h | OSA: 132(122–138) OBA:122.5(112–132) | OSA:5 in 33 OBA:10 in 32 |
| Marcin Wiech 2022 [24] | Poland | OSA:56.4 (51.0–61.8) OBA1:53.1 (46.8–59.4) OBA2:57.1 (50.4–63.8) | OSA:26.7 (24.6–28.9) OBA1:25.3 (22.9–27.7) OBA2:26.5 (24.4–28.5) | Mastectomy or BCS combined with SLNB or ALND | OSA:22 OBA1:22 OBA2:21 | Propofol Sevoflurane Fentanyl | Ultrasound-guided ESPB: 0.375% ropivacaine at 0.4mL/kg (max 40mL). | OBA1: Without nerve block OBA2: Sham nerve block | Oxycodone (PCA: 1mg/dose); paracetamol 1 g IV every 6h. If VAS>4, give oxycodone 5mg IV. | QoR-40 at POD 24 h | OSA: 186(177–193) OBA1: 175 (165–183) OBA2:181 (169–188) | Not recorded |

(Continued)

Table 1. (Continued)

| Study | Country | Age | BMI | Surgery | Sample | Same strategy | Experimental group strategy | Control group strategy | Postoperative medication | Recovery scale | Results | PONV24 |
|---|---|---|---|---|---|---|---|---|---|---|---|---|
| Yao Y 2019 [25] | China | OSA:51.2±5.3 OBA:52.9±4.8 | OSA:23.3±2.6 OBA:22.9±2.1 | Modified Radical Mastectomy | OSA:39 OBA:40 | Propofol Sufentanil Sevoflurane | Ultrasound-guided ESPB: 25mL of 0.5% ropivacaine. | Sham nerve block | Sufentanil PCIA with a basal infusion rate of 1 µg/h and a bolus dose of 2 µg, combined with flurbiprofen axetil 50mg IV every 8 hours. | QoR-15 at POD 24 h | OSA: 120(118–124) OBA: 110(108113) | OSA:3 in 39 OBA:9 in 40 |
| Yusheng Yao 2019 [26] | China | OSA:46.5±10.4 OBA:47.7±9.8 | OSA:22.1±3.4 OBA:21.7±2.0 | Mastectomy or BCS combined with SLNB or ALND | OSA:34 OBA:34 | Propofol Sufentanil Sevoflurane | Ultrasound-guided SPB: 25ml of 0.5% ropivacaine. | Sham nerve block | Sufentanil PCIA: background rate 2 µg/h, bolus 2 µg. Flurbiprofen axetil 50mg IV every 12h | QoR-40 at POD 24 h | OSA: 158(154–159) OBA: 141(139–145) | OSA:2 in 34 OBA:9 in 34 |
| S. Hontoir 2016 [27] | Belgium | OFA:53.9±14.5 OBA:53.2±11.1 | OFA:27.1±4.6 OBA:26.4±4.9 | Mastectomy or BCS combined with ALND | OFA:31 OBA:32 | Propofol Lidocaine Ketamine Sevoflurane | Induction: clonidine (0.2 µg/kg loading dose), ketamine (0.3mg/kg), and lidocaine (1.5mg/kg). Maintenance: additional ketamine (0.2mg/kg) when necessary. | Induction: remifentanil TCI, ketamine 0.3mg/kg, lidocaine 1.5mg/kg, propofol. Maintenance: remifentanil TCI | Intravenous paracetamol 1000mg every 6 hours and intravenous diclofenac 75mg every 12 hours for 24 hours; when VAS≥4, administer PCIA with pethidine 2mg per dose. | QoR-40 at POD 24 h | OFA:182.1±13.9 OBA:175.6±14.8 | Not recorded |
| Jiawei Chen 2025 [28] | China | OFA:46.4±7.5 OSA:48.0±8.6 | OFA:23.1±2.3 OSA:23.6±3.3 | Mastectomy or BCS combined with SLNB or ALND | OFA:65 OSA:67 | PECS Propofol | Before intubation, dexmedetomidine 1 µg/kg IV infusion over 10min, followed by maintenance at 0.5–1.5 µg/kg/h. | Induction: sufentanil 0.5 µg/kg IV. Intraoperative: sufentanil 0.025 µg/kg as needed. | Sufentanil PCIA 2 µg per bolus dose combined with parecoxib 40mg IV every 12 hours | QoR-15 at POD 24 h | OFA:126.9±10.3 OSA:127.4±14.0 | OFA:0 in 65 OSA:4 in 68 |
| Bengu G 2025 [29] | Turkey | OSA:55.3±10.2 OBA:53.7±10.1 | OSA:28.7±4.6 OBA:29.3±5.7 | Modified Radical Mastectomy | OSA:30 OBA:30 | Sevoflurane Remifentanil Propofol Lidocaine Fentanyl | Ultrasound-guided superior posterior serratus intercostal plane block: 20mL of 0.25% bupivacaine. | Without nerve block | Paracetamol 10mg/kg every 8h. PCA: tramadol 10mg bolus, for NRS≥4, give fentanyl 25 µg IV; if ineffective, IM diclofenac 75mg then IV pethidine 0.25mg/kg. | QoR-15 at POD 24 h | OSA:135(105–150) OBA:124.5(82–142) | OSA:9 in 30 OBA:20 in 30 |

(Continued)

**Table 1.** (Continued)

| Study | Country | Age | BMI | Surgery | Sample | Same strategy | Experimental group strategy | Control group strategy | Postoperative medication | Recovery scale | Results | PONV24 |
|---|---|---|---|---|---|---|---|---|---|---|---|---|
| **Gokcen Kulturoglu 2024** [30] | Turkey | OSA1:53±9.4 OSA2:51.9±8.5 OBA:52.6±8.5 | OSA1:28.1±6.1 OSA2:28±5.4 OBA:29.5±10.5 | Modified Radical Mastectomy | OSA1:24 OSA2: 24 OBA:24 | Sevoflurane Remifentanil Propofol Lidocaine Fentanyl | OSA1: Ultrasound-guided rhomboid intercostal nerve block: 30 mL of 0.25% bupivacaine. OSA2: Ultrasound-guided pectoral nerve block: 30 mL of 0.25% bupivacaine. | Without nerve block | Intravenous paracetamol 1 g every 8h+tramadol PCA (20 mg bolus); for VAS≥4, give dexketoprofen 50 mg IV. | QoR-40 at POD 24 h | OSA1: 183±8.8 OSA2: 184.5±8.2 OBA: 179.3±7.1 | OSA1:2 in 24 OSA2:2 in 24 OBA:6 in 24 |
| **Md Hammad Mohsin 2023** [31] | India | OSA1:43.52±9.6 OSA2: 42.26±7.6 OBA:44.13±8.9 | OSA1:22.5±3.8 OSA2:23.9±3.6 OBA:22.0±3.1 | Modified Radical Mastectomy | OSA1:30 OSA2: 30 OBA:30 | Propofol Remifentanil Fentanyl | OSA1: Ultrasound-guided PECS. PECI block involves injecting 20 ml of 0.25% bupivacaine between the pectoralis major and minor muscles, while PECII block administers 20 ml of 0.25% bupivacaine between the pectoralis minor and serratus anterior muscles. OSA2: Ultrasound-guided erector spinae plane block with 20 ml of 0.25% bupivacaine. | Without nerve block | Intravenous paracetamol 1 g every 6h; when VAS >3, administer tramadol 100 mg IV. | QoR-40 at POD 24 h | OSA1: 183.64±6.36 OSA2: 179.68±6.38 OBA: 171.37±6.88 | Not recorded |

Breast-conserving Surgery (BCS), Sentinel Lymph Node Biopsy (SLNB), Axillary Lymph Node Dissection (ALND), postoperative day (POD), deep serratus anterior plane block (D-SAPB), serratus plane block (SPB), thoracic paravertebral block (TPVB), patient-controlled intravenous analgesia (PCIA), modified intercostal nerve block (MINB), erector spinae plane block (ESPB), target-controlled infusion (TCI), pectoral nerve block (PECS)

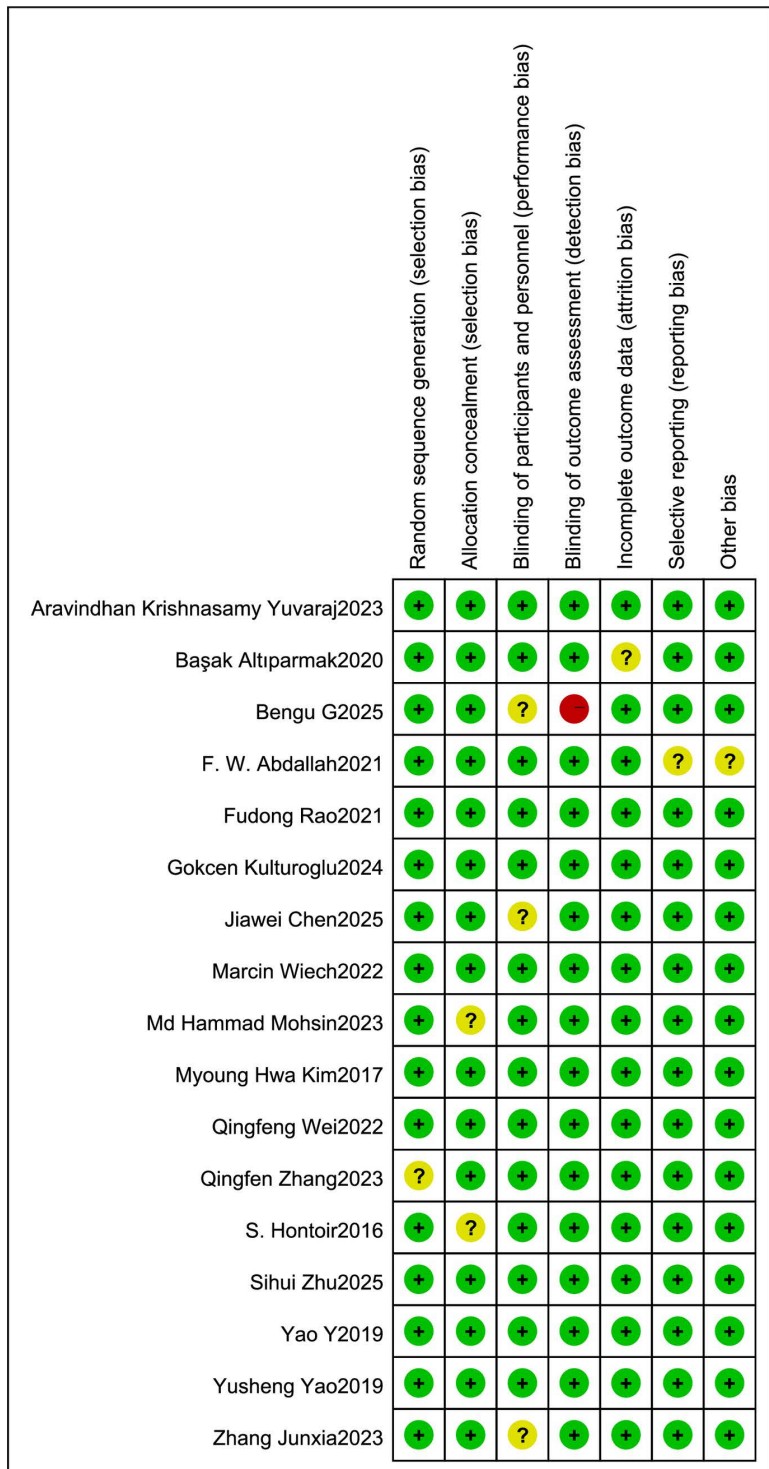

**Fig 2. Risk of bias assessment of the included studies.**

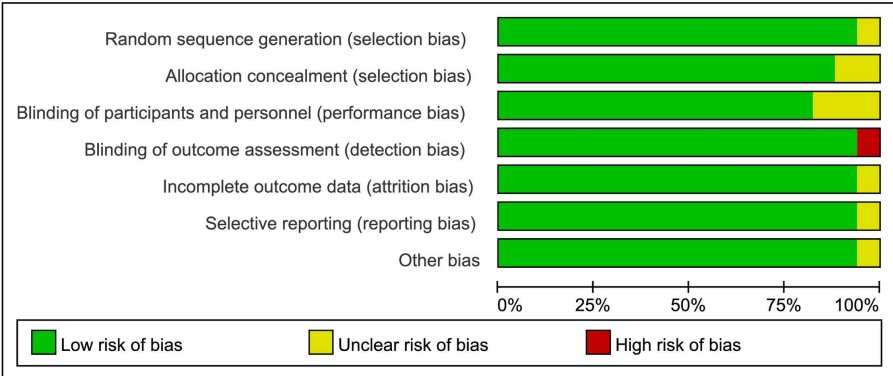

**Fig 3. Percentage of projects that introduced a risk of bias among the included RCTs.**

## Quality of evidence assessment

The credibility of evidence for each outcome in the network was evaluated using the GRADE framework. The evidence for the primary outcome of OSA versus OBA, as well as the incidence of PONV at 24 hours in the OSA versus OBA comparison, was rated as high quality. All other outcomes were assessed as having moderate to low quality. Detailed information is provided in Supplementary Material (S3 Text).

## Model quality assessment and network diagram

This study evaluated the model convergence, sampling efficiency, and inconsistency of both primary and secondary outcomes. The results showed that all Gelman-Rubin diagnostic Rhat values were <1.05, indicating good convergence of the MCMC chains. The effective sample size for all parameters exceeded 1,000, meeting the reliability criteria for posterior inference. Local inconsistency was assessed using the node-splitting method, and all comparisons yielded P-values >0.05, supporting the consistency of the network. The specific results are provided in the supplementary materials (S5 Text).

The network diagrams for the 24-hour postoperative QoR scores and their subscale scores are presented in Fig 4. In the network plots, each node represents an intervention, with the number above indicating the total number of patients receiving that intervention. Lines between nodes represent direct comparisons between interventions, with thicker lines indicating a greater number of direct comparison studies.

## Assessment of postoperative quality of recovery at 24 hours

Seventeen studies involving a total of 1,254 patients reported 24-hour postoperative QoR scores. Based on the Deviance Information Criterion (DIC), a random-effects model was used for effect size estimation. The NMA results indicated that, compared with the OBA group, OFA significantly improved postoperative recovery quality (mean difference [d] = 0.044, 95% credible interval [CrI]: 0.020–0.068), and OSA demonstrated an even greater improvement (d = 0.050, 95% CrI: 0.038–0.062). The comparison between OFA and OSA showed no statistically significant difference (d = −0.006, 95% CrI: −0.029 to 0.018).Between-study heterogeneity was low (sd.d = 0.018), and the Bayesian $\tau^2$-transformed $I^2$ statistic was 0%, supporting the homogeneity of the findings. Detailed results are presented in the forest plot (For all result forest plot, refer to Fig 5).

Regarding postoperative recovery quality in breast cancer patients, the Surface Under the Cumulative Ranking (SUCRA) analysis ranked OSA highest (85.3%), followed by OFA (64.7%), and OBA (0.0%). The cumulative ranking

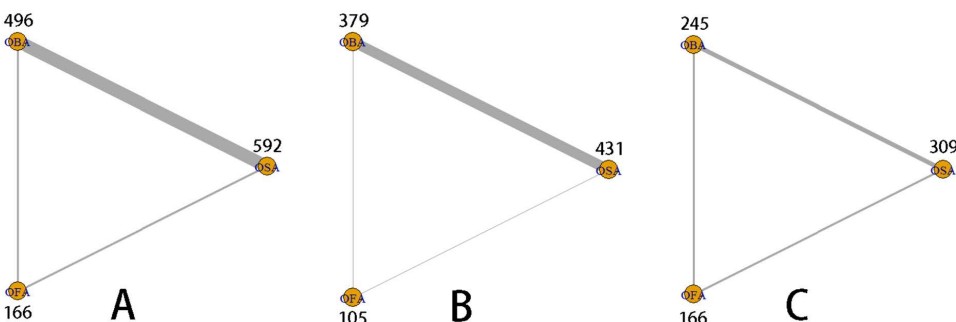

**Fig 4. Network evidence.** The postoperative 24-hour QOR network plot (A). The postoperative 24-hour PONV network plot (B). The postoperative 24-hour QOR subscale network plot (C).

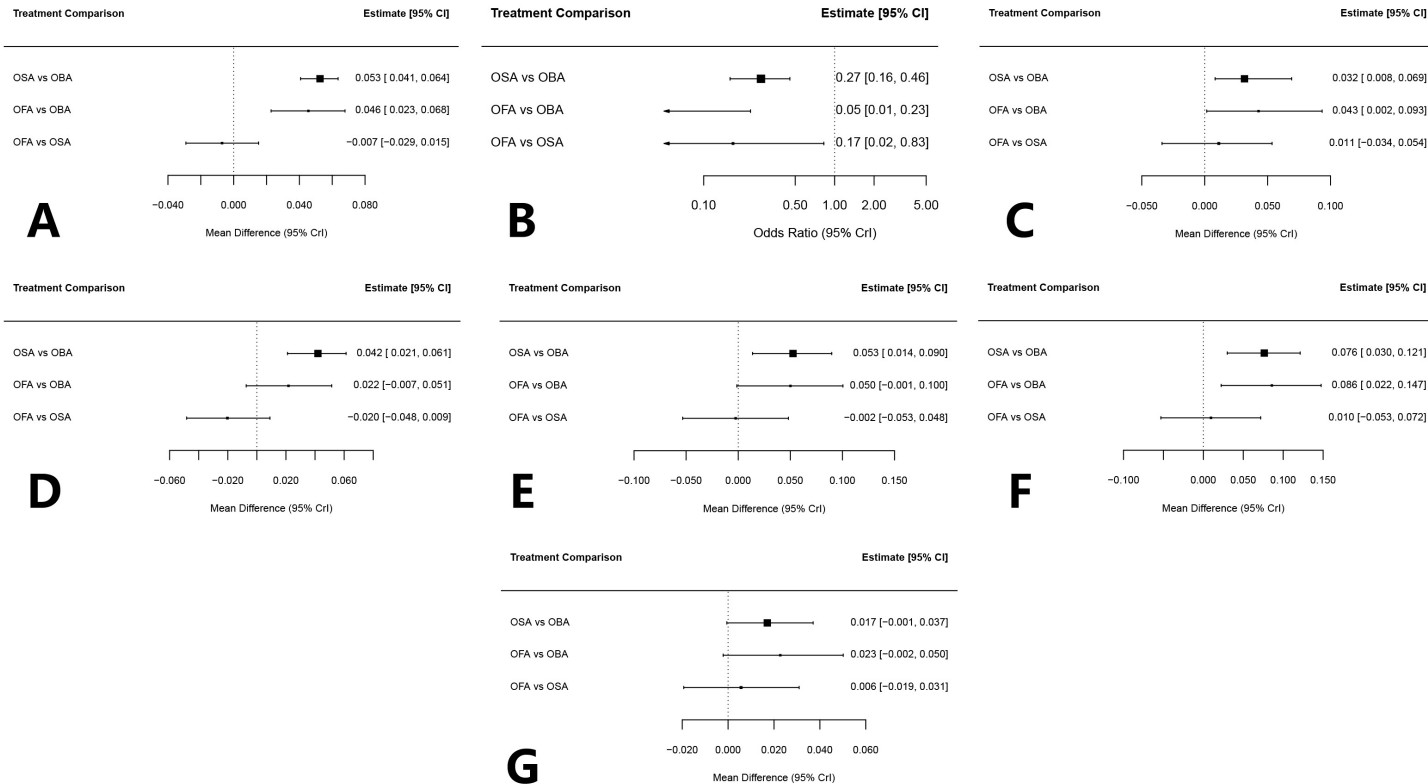

**Fig 5. Forest plot.** QOR score (A), PONV (B), physical independence (C), emotional state (D), physical comfort (E), pain (F), psychological support (G).

probability curve further confirmed that OSA had a consistently favorable ranking in improving postoperative recovery (See Fig 6 for all result details).

## Assessment of postoperative nausea and vomiting at 24 hours

Eleven studies involving a total of 779 patients reported on the incidence of PONV within 24 hours after surgery. Based on comparison of DIC values, which showed minimal differences, a random-effects model was applied for analysis. The

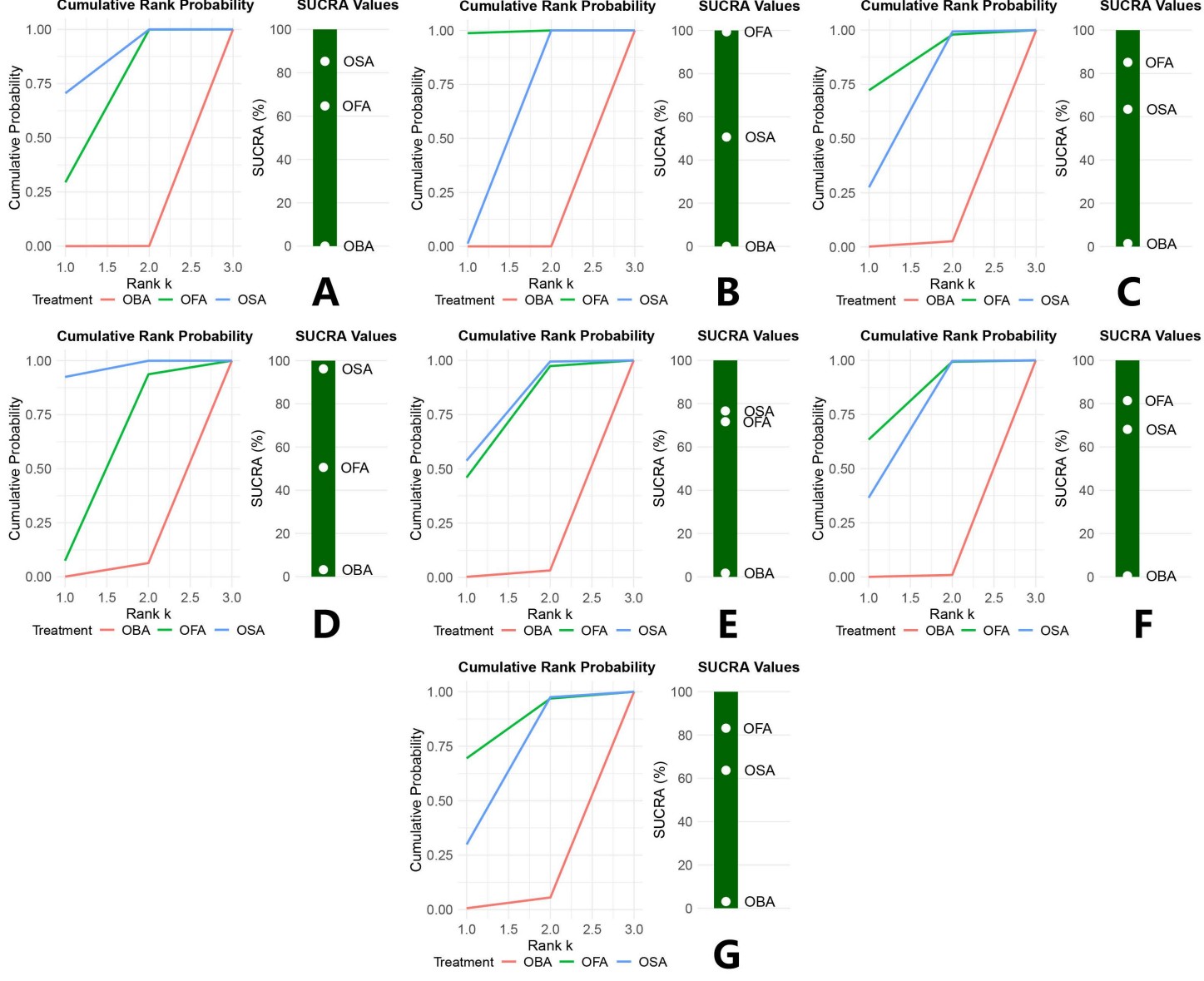

**Fig 6. Cumulative ranking probability and SUCRA plot.** QOR score (A), PONV (B), physical independence (C), emotional state (D), physical comfort (E), pain (F), psychological support (G).

NMA results demonstrated that, compared to OBA, both OSA (OR = 0.27, 95% CrI: 0.16–0.46) and OFA (OR = 0.05, 95% CrI: 0.01–0.22) significantly reduced the incidence of 24-hour postoperative PONV, with statistically significant differences. Moreover, OFA resulted in fewer PONV events than OSA, also with a statistically significant difference (OR = 0.17, 95% CrI: 0.02–0.83).Heterogeneity across studies was low (sd.d = 0.273, I² = 0%), indicating consistency in the findings.

Regarding PONV outcomes in breast cancer surgery patients, the SUCRA analysis showed OFA had the highest probability of being the most effective (99.4%), followed by OSA (50.6%), and OBA (0.0%). The cumulative ranking probability curve confirmed OFA's consistent superiority in reducing PONV incidence.

## Assessment of QoR subscale scores at 24 hours postoperatively

Nine studies involving a total of 720 patients reported outcomes from the subscales of the QoR scale. Model comparison based on the DIC showed that random-effects models provided a better fit for all subscales except emotional state; however, the DIC difference (ΔDIC) for the emotional state subscale was < 3, indicating minimal difference. Therefore, random-effects models were applied uniformly across all subscales.

For the physical independence subscale, the NMA indicated that both OSA and OFA significantly improved physical independence compared to OBA (OSA: d = 0.032, 95% CrI: 0.008–0.069; OFA: d = 0.043, 95% CrI: 0.002–0.093). The comparison between OFA and OSA did not show a statistically significant difference (d = 0.011, 95% CrI: –0.034 to 0.053). Heterogeneity was low (sd.d = 0.022, $I^2$ = 14%).

SUCRA analysis and cumulative ranking probability curve indicated that, in terms of physical independence recovery after breast cancer surgery, OFA had the highest probability of being the most effective intervention (85.1%), followed by OSA (63.5%) and OBA (1.4%), demonstrating OFA's consistent advantage in this domain.

The NMA results for the emotional state subscale showed that, compared with OBA, both OSA (d = 0.042, 95% CrI: 0.021–0.061) and OFA (d = 0.022, 95% CrI: –0.007–0.051) had positive effects on emotional recovery, although only OSA demonstrated a statistically significant difference. There was no significant difference between OFA and OSA (d = –0.020, 95% CrI: –0.048–0.009). Heterogeneity was low (sd.d = 0.016, $I^2$ = 3%).

SUCRA analysis and cumulative ranking probability curves indicated that, for emotional state recovery in breast cancer surgery patients, OSA had the highest probability of being the optimal intervention (96.2%), followed by OFA (50.6%) and OBA (3.2%), demonstrating OSA's consistent advantage in improving postoperative emotional status.

The NMA results for the physical comfort subscale indicated that, compared with OBA, both OSA (d = 0.053, 95% CrI: 0.014–0.090) and OFA (d = 0.050, 95% CrI: –0.001–0.100) improved physical comfort, although only the effect of OSA was statistically significant. No significant difference was observed between OFA and OSA (d = –0.002, 95% CrI: –0.053–0.048). Heterogeneity was low (sd.d = 0.041, $I^2$ = 4%).

SUCRA analysis and cumulative ranking probability curves showed that, in terms of postoperative physical comfort in breast cancer surgery patients, OSA had the highest probability of being the optimal intervention (76.6%), followed by OFA (71.7%) and OBA (1.7%), indicating a slight advantage for OSA in this domain.

The NMA results for the pain subscale showed that, compared with OBA, both OSA (d = 0.076, 95% CrI: 0.030–0.121) and OFA (d = 0.086, 95% CrI: –0.022–0.148) were effective in improving pain-related recovery, with both effects reaching statistical significance. There was no significant difference between OFA and OSA (d = 0.010, 95% CrI: –0.053–0.072). Heterogeneity was low (sd.d = 0.049, $I^2$ = 5%).

SUCRA analysis and cumulative ranking probability curves indicated that, for postoperative pain in breast cancer surgery patients, OFA had the highest probability of being the optimal intervention (81.4%), followed by OSA (68.1%) and OBA (0.5%), suggesting a slight advantage of OFA in alleviating postoperative pain.

The NMA results for the psychological support subscale indicated that none of the comparisons among the three interventions showed statistically significant differences: OSA vs. OBA (d = 0.017, 95% CrI: –0.001 to 0.037), OFA vs. OBA (d = 0.023, 95% CrI: –0.002 to 0.050), and OFA vs. OSA (d = 0.005, 95% CrI: –0.019 to 0.031). Heterogeneity was low (sd.d = 0.016, $I^2$ = 12%).

SUCRA analysis and cumulative ranking probability curves showed that, in terms of psychological support for breast cancer surgery patients, OFA had the highest probability of being the most effective intervention (83.2%), followed by OSA (63.7%) and OBA (3.1%), indicating a slight advantage for OFA in this domain.

## Dose–response relationship

A total of 11 studies involving 753 participants reported intraoperative opioid use. A meta-regression was conducted to examine the association between the percentage reduction in opioid use and improvement in QoR scores at 24 hours

postoperatively. The test for linearity showed no significant linear association between the percentage reduction in opioid use and QoR improvement (β = 0.0001, 95% CI: –0.0002 to 0.0004, p = 0.601), with the linear model accounting for 0% of the heterogeneity ($R^2$ = 0%).A test for nonlinearity using restricted cubic splines (with 3 degrees of freedom) revealed a significant nonlinear association between the reduction in opioid use and QoR score improvement (QM = 18.01, df = 3, p = 0.0004). The spline model explained 100% of between-study heterogeneity ($R^2$ = 100%), significantly outperforming the linear model (likelihood ratio test: LRT = 7.65, p = 0.022, ΔAIC = 3.65). The dose–response curve indicated an inverted U-shaped nonlinear relationship: as the percentage reduction in opioid use increased, the QoR score initially declined, then rose, and subsequently declined again. This pattern suggests the presence of an optimal range for opioid reduction. Fig 7 illustrates the dose–response nonlinear trend and residual patterns.

### Sensitivity analysis

Sensitivity analyses were conducted for all outcomes by comparing prior and posterior distributions of heterogeneity and treatment effects. For heterogeneity, half-normal and inverse-gamma priors were applied; for treatment effects, a half-Cauchy prior was used. The resulting effect sizes and DIC values showed minimal differences across these priors. Detailed results are presented in the supplementary materials (S6 Text).

To assess the robustness of the findings, sensitivity analyses were performed by excluding one study with a high risk of bias and four studies involving multi-arm comparisons. The direction of effect estimates remained consistent across all outcomes that included these studies.

Additionally, to evaluate the impact of scale transformation, the Bayesian network meta-analysis was repeated using standardized mean differences (SMD) as effect sizes. The sensitivity results differed slightly from the primary analysis: OSA vs. OBA (d = 1.286, 95% CrI: 0.700–1.869), OFA vs. OBA (d = 0.886, 95% CrI: –0.232–2.016), and OFA vs. OSA (d = –0.400, 95% CrI: –1.529–0.711). Treatment ranking remained largely consistent with SUCRA values: OSA (88.6%)> OFA (58.6%)> OBA (2.8%). Detailed sensitivity analysis results are provided in the supplementary materials (S7 Text).

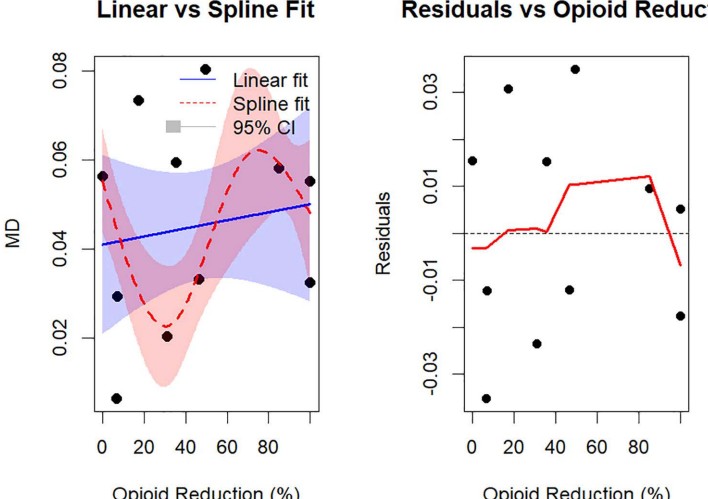

**Fig 7.  Dose–response nonlinearity and residual diagnostics.**

## Publication bias

Funnel plots were generated for all outcomes, and Egger's regression tests were performed to assess potential publication bias. The results indicated that, except for the physical independence subscale, all funnel plots demonstrated good symmetry, with Egger's test yielding p-values > 0.05, suggesting a low likelihood of publication bias. However, the funnel plot for the physical independence subscale showed poor symmetry, indicating a potential presence of publication bias or small-study effects for this outcome. Funnel plots and the results of Egger's regression tests are presented in the supplementary materials (S4 Text).

## Discussion

This study is the first to use breast cancer surgery as an ideal model to systematically compare the comprehensive effects of OFA, OSA, and OBA on early postoperative recovery quality through Bayesian network meta-analysis. Earlier, the Yijun Liu team [32] attempted to explore the effect of OFA on QoR using traditional meta-analysis, but unfortunately, due to the inclusion of mixed types of surgeries and varying anesthetic protocols, the conclusion exhibited significant heterogeneity ($I^2 > 90\%$), and the quality of evidence was limited, making it difficult to guide clinical decision-making. In this study, by restricting the surgical types, we significantly reduced the heterogeneity of the primary outcomes (sd.d = 0.018, $I^2 = 0\%$) and leveraged the characteristics of breast cancer surgery, which serves as an ideal model for exploring the value of OFA, avoiding the risks associated with OFA in more complex major surgeries. These improvements have notably enhanced the reliability of the results and provide high-quality evidence for clinical decision-making.

The main findings of this study indicate that OSA demonstrates significant advantages in the comprehensive assessment of recovery quality within 24 hours postoperatively. The effect size of OSA compared to OBA (d = 0.050, 95% CrI 0.038–0.062), when converted according to the minimum clinically important difference (MCID) of 6.3 points for the QoR-40 scale [33], requires the confidence interval to exceed 0.0315—thus clearly reaching the threshold for clinical significance. When calculated using the MCID of 6 points for the QoR-15 scale [34] (corresponding to d ≥ 0.04), the improvement from OSA is close to, but does not entirely reach, clinical significance. Considering that the evidence from the QoR-40 scale is more robust (9 studies, 658 patients vs. 8 studies, 596 patients for QoR-15), and that QoR-40 provides greater measurement precision as a more comprehensive tool, the improvement seen with OSA over OBA can be considered clinically meaningful.In contrast, the effect size of OFA compared with OBA did not surpass the MCID threshold. This may relate to findings by Evan D [35], who reported that OFA is associated with inadequate analgesia and increased safety risks. Overemphasis on "opioid-free" strategies may overlook the importance of individualized balance and multimodal analgesia. It is hypothesized that, in order to compensate for reduced analgesic potency, OFA often requires increased doses of non-opioid medications, which may in turn exacerbate side effects and slightly impair recovery scores. In the OSA regimen, small doses of opioids help attenuate the surgical stress response and maintain hemodynamic stability, avoiding the adverse reactions caused by high doses of non-opioid agents. Mechanistically, the superiority of OSA reflects the core logic of precision anesthesia—using the minimum effective dose of opioids to block key pathophysiological pathways, while maximizing the synergistic effects of non-opioid techniques. This strategy avoids the compensatory medication burden seen in completely opioid-free protocols and aligns more closely with real-world clinical practice.

Secondary outcomes revealed that although OFA demonstrated an absolute advantage in controlling PONV, it lagged behind OSA in overall physical comfort assessment. This seemingly paradoxical result can be explained by the multidimensional nature of the physical comfort subscale. While PONV is an independent measure reflecting gastrointestinal symptoms such as nausea, vomiting, and retching, the physical comfort subscale encompasses not only these symptoms but also non-gastrointestinal factors such as shivering, dizziness, trembling, feeling cold, and sleep quality. This study hypothesizes that to ensure adequate analgesia, the OFA protocol frequently employs α2-adrenergic agonists, NMDA antagonists, and ion channel blockers. Although these agents may alleviate PONV,α2 receptor agonists can inhibit the sympathetic nervous system, thereby diminishing its thermoregulatory effects. This leads to a decrease in the patient's

body temperature, consequently increasing subjective discomfort [36,37]. NMDA receptor blockade may weaken GABAergic neuronal inhibition, resulting in agitation and dizziness [38], while the narrow therapeutic window of ion channel blockers [39] increases the risk of dizziness and perioral numbness, which can interfere with eating. The interplay of these adverse effects may offset OFA's advantage in PONV control.

In postoperative pain management, the OFA strategy demonstrates greater advantages compared to OSA, likely due to its avoidance of opioid-induced hyperalgesia. Opioids can activate the spinal dorsal horn NMDA receptor pathway via rapid μ-receptor internalization, thereby exacerbating postoperative pain hypersensitivity [40]. OFA, by entirely eliminating opioid exposure and utilizing ketamine's inhibitory effect on central sensitization, effectively blocks this mechanism. Regarding physical independence, both OFA and OSA are significantly superior to OBA, suggesting enhanced support for early mobilization. The outstanding performance of OFA may be attributed to synergistic effects such as reducing PONV to minimize reluctance to move, utilizing non-opioid agents to suppress inflammatory movement-related pain, reducing the incidence of hyperalgesia, and avoiding opioid-induced dizziness, all of which contribute to improved postoperative mobility.In terms of emotional status, OSA significantly outperforms OFA, possibly due to the neuroaffective regulatory effects of small doses of opioids, including stress reduction mediated by κ-receptors in the amygdala and a calming effect from μ-receptor-activated dopamine release in the nucleus accumbens [41]. The emotional outcomes observed with OFA may be linked to the rapid antidepressant effect of ketamine, which works by blocking NMDA receptors and activating the AMPA/mTOR pathway to promote the release of BDNF in the prefrontal cortex, thereby supporting emotional stability [42]. For psychological support, no statistically significant differences were observed among the three groups. This dimension is primarily influenced by non-pharmacological factors, such as the quality of physician-patient communication, and is thus minimally affected by anesthesia strategies.In summary, although complete opioid avoidance offers significant benefits in reducing PONV and pain, its advantages may be offset by compounded side effects from adjunct medications, ultimately leading to a decline in overall recovery scores. This finding aligns with Blum's OFA concept [43], which emphasizes that the core goal of OFA is to minimize—not rigidly eliminate—opioid exposure. Its clinical value lies in achieving individualized recovery through minimized opioid-related risk rather than absolute opioid-free anesthesia. Therefore, this study suggests that OFA should be viewed as a deeper implementation of the OSA approach. The ultimate goal is to minimize intraoperative opioid use, but successful clinical implementation depends on selecting the most appropriate regimen based on the patient's individual characteristics, the type of surgery, and the available techniques, in order to best reduce opioid-related risks. Rational reduction in opioid use is key to improving postoperative recovery quality.

The dose–response analysis further supports the above discussion, revealing an inverted U-shaped relationship between the percentage reduction in opioid use and improvement in postoperative QoR scores. Specifically, increasing the percentage reduction in opioid use within a certain range enhances QoR; however, beyond a certain threshold, further opioid reduction may paradoxically impair recovery quality. This nonlinear trend suggests the existence of an optimal opioid reduction range—likely between 40% and 80%—within which patients experience the greatest recovery benefit. Nevertheless, this finding should be interpreted with caution due to several limitations: the included studies employed highly heterogeneous intervention strategies, postoperative analgesia protocols were not standardized, sample sizes were limited, data were sparse in the high reduction range, and there was a lack of medium- to long-term recovery data. These factors hinder the ability of opioid reduction percentage alone to independently predict postoperative QoR, thereby limiting the robustness of the conclusion.

To assess the robustness of the linear transformation method—used to rescale QoR-15 and QoR-40 scores to a 0–1 interval for pooling—the study repeated the Bayesian network meta-analysis using SMD as the effect measure. The comparison between OFA and OBA, which was significant in the primary analysis, became non-significant under SMD. This shift is primarily attributed to the inherent limitations of SMD: (1) substantial heterogeneity in standard deviations across studies can markedly widen the confidence interval, increasing the likelihood of the effect estimate crossing zero; (2) SMD is highly sensitive to variability in standard deviation estimation, potentially amplifying study bias; and (3) its

clinical interpretability is abstract, easily distorted by extreme values in standard deviations, and may not accurately reflect benefits on the original scale.Nonetheless, these results reinforce the stability of the main conclusions: the key comparisons—OSA vs OBA and OSA vs OFA—retained both statistical significance and directional consistency. Furthermore, the SUCRA rankings were highly consistent between the primary and SMD-based analyses, with OSA consistently ranked as the most favorable strategy. These findings also support the superiority of the linear transformation method adopted in the primary analysis: it offers greater stability, is less sensitive to differences in standard deviation, and allows for more intuitive clinical interpretation through percentage improvement scores. Additionally, the structural similarity between QoR-15 and QoR-40, along with their strong score correlation ($R^2 \approx 0.97$) [44], further justifies linear transformation. Therefore, for highly homologous scales, the linear transformation method—by preserving the clinical interpretability of the original scoring units—represents a more suitable approach for pooled analysis in clinical decision-making.

This study has several limitations: (1) The number of included RCTs is relatively limited. While this does not compromise the robustness of the primary conclusions, it imposes constraints when exploring the nonlinear dose–response relationship between opioid reduction percentage and QoR; (2) There is considerable heterogeneity in intervention strategies. The specific methods of opioid reduction in OSA and OFA vary significantly;moreover, the OBA medication regimens vary, resulting in differences in baseline opioid consumption, increasing between-study heterogeneity and limiting the predictive utility of opioid reduction percentage as an independent indicator of QoR; (3) The surgical procedures are not unified and postoperative analgesic regimens are not standardized, and their variations may confound both postoperative recovery quality and the potential benefits of intraoperative opioid reduction; (4) The study primarily focuses on short-term outcomes within 24 hours postoperatively, lacking medium- and long-term recovery data; (5) While the statistical dependence issue in four multi-arm RCTs was mitigated by merging subgroups, this approach may obscure differences between subgroups and influence weighting of sample sizes.Despite these limitations, the robustness of the primary findings is supported by the enhanced effect sizes for OSA (from 0.050 to 0.060) and OFA (from 0.044 to 0.050) compared with OBA after exclusion of multi-arm merged studies. This affirms the clinical value of reducing intraoperative opioid exposure and suggests that optimizing recovery quality depends on tailoring intervention combinations to individual patient needs rather than completely eliminating opioids. Future research should be conducted within a standardized perioperative management framework to evaluate the effects of various non-opioid strategies on different QoR dimensions to identify the optimal anesthetic approach and verify the applicability of the optimal opioid reduction range through larger sample sizes and long-term follow-up, thereby guiding individualized treatment planning.

## Conclusion

This study demonstrates that OSA has significant clinical value in enhancing the overall quality of recovery following breast cancer surgery, making it a preferred strategy for optimizing postoperative rehabilitation. OFA also shows excellent efficacy in controlling postoperative nausea, vomiting, and pain management. Both approaches provide effective anesthetic strategies for recovery after breast cancer surgery. However, complete opioid elimination is not necessarily superior; clinical practice should emphasize precision anesthesia and individualized medication to optimize therapeutic outcomes.

## Supporting information

**S1 Table.  Prisma checklist.** Prisma 2020 check list.
(DOCX)

**S1 Text.  Search strategy.** Search terms and database strategies.
(DOCX)

**S2 Text.  Full-text exclusion reasons.** The reasons for exclusion after full-text review.
(DOCX)

**S3 Text. Quality of evidence assessment.** GRADE evaluation for primary/secondary outcomes.
(DOCX)

**S4 Text. Publication bias results.** Funnel plots and the results of Egger's regression tests.
(DOCX)

**S5 Text. Model diagnostics.** Results of Gelman-Rubin diagnostic, effective sample size, and Node-Splitting inconsistency test.
(DOCX)

**S6 Text. Heterogeneity & effect comparisons.** Heterogeneity and comparison of prior and posterior treatment effects.
(DOCX)

**S7 Text. Sensitivity analyses.** Results of sensitivity tests.
(DOCX)

**S8 Text. Bayesian sequential meta-analysis.** Stability of findings via cumulative evidence.
(DOCX)

## Author contributions

**Conceptualization:** JingWang Liu.

**Data curation:** JingWang Liu, MingJie Wang.

**Formal analysis:** JingWang Liu, JiaXin Liu.

**Funding acquisition:** XiuLi Wang.

**Investigation:** JingWang Liu, MingJie Wang.

**Methodology:** JingWang Liu, JiaXin Liu, MingJie Wang, XiuLi Wang.

**Project administration:** XiuLi Wang.

**Resources:** XiuLi Wang.

**Software:** JingWang Liu, JiaXin Liu, MingJie Wang, XiuLi Wang.

**Supervision:** XiuLi Wang.

**Validation:** JingWang Liu, MingJie Wang.

**Visualization:** JingWang Liu, JiaXin Liu, MingJie Wang.

**Writing – original draft:** JingWang Liu, JiaXin Liu.

**Writing – review & editing:** JingWang Liu, JiaXin Liu.

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
