## [Decision Letter · Decision Letter 0]

24 Jul 2025

Dear Dr. Wang,

Thank you for submitting your manuscript to PLOS ONE. After careful consideration, we feel that it has merit but does not fully meet PLOS ONE’s publication criteria as it currently stands. Therefore, we invite you to submit a revised version of the manuscript that addresses the points raised during the review process.

We look forward to receiving your revised manuscript.

Kind regards,

Benjamin Benzon, Ph.D., M.D.

Academic Editor

PLOS ONE

Journal Requirements:

4. Please upload a copy of Figure 1-7, to which you refer in your text on page 7, 8, 11, 14 and 15. If the figure is no longer to be included as part of the submission please remove all reference to it within the text.

**Additional Editor Comments:**

The reviewers made some comments, I am looking forward to you response.

Reviewers' comments:

Reviewer's Responses to Questions

**Comments to the Author**

1. Is the manuscript technically sound, and do the data support the conclusions?

Reviewer #1: Partly

Reviewer #2: Yes

Reviewer #3: Yes

2. Has the statistical analysis been performed appropriately and rigorously?

Reviewer #1: Yes

Reviewer #2: Yes

Reviewer #3: Yes

3. Have the authors made all data underlying the findings in their manuscript fully available?

Reviewer #1: Yes

Reviewer #2: Yes

Reviewer #3: Yes

4. Is the manuscript presented in an intelligible fashion and written in standard English?

Reviewer #1: Yes

Reviewer #2: Yes

Reviewer #3: Yes

**Reviewer #1:**  I am concerned regarding QoR. Although QoR-40 and QoR-15 are evaluating self reported quality of recovery in the last 24 days the instrument are not same. The QoR-40 is having 40 items thet are sorted in five dimensions. Although there are five dimensions in QoR-40, the QoR-15 does not have five dimensions. The QoR-15 was developed from 16 items from QoR-40, and items have different likerrt scale (11) and QoR-40 is having 5. Although there are both used, it is not clear is better shorter and more simple QoR-15 or more comprehensive but time consuming QoR-40.

I think both instruments are great way for evaluation QoR but they are not comparable.

Can you explain to readers why did you decide to compare different but related QoR instruments in this study and why is it not good.

**Reviewer #2:**  Dear Authors,

I really enjoyed reading your review.

I don't completely agree with the notion presented in line 388 that alpha 2 antagonist could induce hypothermia. I believe that contribution to hypothermia is minor (reducing body temp by 0.5 degrees) and does not contribute to significant postoperative hypothermia (although it is not very enjoyable after an hour or more in cold OR). Also, I believe that it is not the same for the postoperative outcome and quality of recovery if we used OSA or OFA for the axillary node extraction and radical mastectomy. That might need further explanation and/or analysis.

**Reviewer #3:**  It would be great if author compare different techniques of both opioid sparing anaesthesia and opioid free anaesthesia and give a reader an recommendation which one to use.

And in opioid based anaesthesia did the patients received only opioid for analgesia or combination with other analgesic agents?

**Do you want your identity to be public for this peer review?** For information about this choice, including consent withdrawal, please see our Privacy Policy

Reviewer #1: No

Reviewer #2: No

Reviewer #3: No

---

## [Author Response · Author response to Decision Letter 1]

29 Jul 2025

Dear Dr. Benzon,

Thanks for providing us with this great opportunity to submit a revised version of our manuscript. In the manuscript revision, we have indicated the removed sections using red text and strikethrough, while the added sections are highlighted in blue. Additionally, we have addressed each of the journal’s requirements as follows:

1. The manuscript has been revised according to the PLOS ONE formatting guidelines.

2. We agree to the data availability policy. As this meta-analysis is a secondary study, no new raw data were generated. All data were derived from previously published studies. Upon acceptance of the manuscript, we will either send the original data from the included studies to you or upload them to GitHub(https://github.com/whenthereis/away.git).

3. The corresponding author’s ORCID has been verified in the submission system.

4. Figures 1–7 have been uploaded to the system in accordance with the journal's requirements.

5. The supplementary materials have been included in the Supporting Information section at the end of the manuscript, and appropriate citations have been added in the main text.

6. No specific references were requested by the other reviewers.

We hope this revised manuscript has addressed your concerns, and look forward to hearing from you.

Sincerely,

The Authors

Reply to Reviewer #1

Dear Reviewers,

Thank you very much for your time involved in reviewing the manuscript. Your concerns are valid; if the correlation between the scales is low, mixed comparisons could indeed affect the reliability of the results. However, according to the original literature on the QoR-15 scale (Development and Psychometric Evaluation of a Postoperative Quality of Recovery Score The QoR-15, DOI: 10.1097/ALN.0b013e318289b84b), the variance explained between the total scores of QoR-15 and QoR-40 is as high as 97%, and the five dimensions of QoR-15 are entirely derived from QoR-40. The clinical discriminative power of each dimension is similar, indicating that the two scales are highly correlated. Therefore, we used a linear transformation method to eliminate the scale differences and confirmed the robustness of the conclusions through sensitivity analysis (SMD method).Additionally, several meta-analyses on postoperative recovery quality have combined the QoR-15 and QoR-40 scales for comparison (Opioid-free anaesthesia and postoperative quality of recovery: a systematic review and meta-analysis with trial sequential analysis, DOI: 10.1016/j.accpm.2024.101453). This includes studies published in top anesthesiology journals such as BJA (The impact of perioperative ketamine or esketamine on the subjective quality of recovery after surgery: a meta-analysis of randomised controlled trials, DOI: 10.1016/j.bja.2024.03.012). These pieces of evidence support the scientific reliability of combining the two scales for comparison.

Sincerely,

The Authors

Reply to Reviewer #2

Dear Reviewers,

Thank you very much for your time involved in reviewing the manuscript and your very encouraging comments on the merits. We had previously overstated the impact of α2 receptor antagonists on temperature reduction, so We have revised the wording to avoid any absolute statements.

Regarding your second question about the influence of surgical method differences on postoperative recovery quality, We acknowledge that such differences will certainly affect the results. We will add a discussion on this limitation in the study. However, since the surgical methods included are relatively fixed—being either breast-conserving surgery or total mastectomy with sentinel lymph node biopsy or axillary lymph node dissection (modified radical mastectomy = total mastectomy with axillary lymph node dissection)—excluding studies that involve multiple surgical approaches would render it impossible to form a closed-loop network for analysis. This would also prevent subgroup analysis. Moreover, if surgical method differences had a significant impact on the results, it would theoretically increase the heterogeneity of the study. However, the primary outcome of our study shows very low heterogeneity. Therefore, we believe that within the range of surgical methods included, variations in surgical approaches do not affect the robustness of our study’s results.

Sincerely,

The Authors

Reply to Reviewer #3

Dear Reviewers,

Thanks for your great suggestion on our manuscript. Your suggestions are highly valuable and offer important clinical insights. We had previously conducted a network meta-analysis exploring the impact of various non-opioid anesthesia strategies on postoperative recovery quality, aiming to identify the optimal non-opioid anesthesia regimen. However, due to the refinement and complexity of the anesthesia strategies, the number of subgroups increased significantly. To ensure sufficient evidence for comparison, We had to broaden the inclusion criteria by expanding the range of surgical procedures. This ultimately resulted in high heterogeneity and low evidence quality. Therefore, we believe that current evidence is insufficient to support a detailed comparison of specific anesthesia strategies in a meta-analysis. Nonetheless, this is a promising direction, and we plan to conduct further analyses once more robust evidence becomes available.

Regarding your second question, with the advancement of modern anesthetic concepts, perioperative analgesic protocols relying solely on opioids have become increasingly uncommon. In our study, most trials categorized under the opioid-based anesthesia group adopted a strategy of "opioid-dominant analgesia supplemented with non-opioid agents" during surgery. Only a small number of studies employed opioids as the sole intraoperative analgesic. If the intra-group variability within the opioid-based anesthesia group had a significant influence on the outcomes, it would theoretically increase the heterogeneity across studies. However, our primary outcomes demonstrated very low heterogeneity, suggesting that this variability did not compromise the robustness of our findings.

Sincerely,

The Authors

---

## [Decision Letter · Decision Letter 1]

1 Oct 2025

Opioid-sparing anesthesia versus opioid-free anesthesia for postoperative recovery quality in breast cancer surgery patients: A systematic review and Bayesian network meta-analysis

PONE-D-25-33203R1

Dear Dr. Wang,

We’re pleased to inform you that your manuscript has been judged scientifically suitable for publication and will be formally accepted for publication once it meets all outstanding technical requirements.

Kind regards,

Benjamin Benzon, Ph.D., M.D.

Academic Editor

PLOS ONE

Additional Editor Comments (optional):

Reviewers' comments:

Reviewer's Responses to Questions

**Comments to the Author**

Reviewer #1: All comments have been addressed

Reviewer #3: All comments have been addressed

2. Is the manuscript technically sound, and do the data support the conclusions?

Reviewer #1: Partly

Reviewer #3: Yes

3. Has the statistical analysis been performed appropriately and rigorously?

Reviewer #1: No

Reviewer #3: Yes

4. Have the authors made all data underlying the findings in their manuscript fully available?

Reviewer #1: Yes

Reviewer #3: Yes

5. Is the manuscript presented in an intelligible fashion and written in standard English?

Reviewer #1: Yes

Reviewer #3: Yes

Reviewer #1: Thank you for your answers .

My intention was that you explain readers why is ok to combine similar but different questioners.

Can you explain and discuss your response in a work, not just to me.

Although questioners QoR-40 and QoR-15 are correlated and collinear it is not proof that they are similar. Correlation and covariance are not evidence of similarity.

Can you use Bland Altman statistic to show how 2 instruments are similar.

What is with normality and homoscedasticity of QoR-40 and QoR15 score ?

Dear Reviewers,

Thank you very much for your time involved in reviewing the manuscript. Your concerns are valid; if the correlation between the scales is low, mixed comparisons could indeed affect the reliability of the results. However, according to the original literature on the QoR-15 scale (Development and Psychometric Evaluation of a Postoperative Quality of Recovery Score The QoR-15, DOI: 10.1097/ALN.0b013e318289b84b), the variance explained between the total scores of QoR-15 and QoR-40 is as high as 97%, and the five dimensions of QoR-15 are entirely derived from QoR-40. The clinical discriminative power of each dimension is similar, indicating that the two scales are highly correlated. Therefore, we used a linear transformation method to eliminate the scale differences and confirmed the robustness of the conclusions through sensitivity analysis (SMD method).Additionally, several meta-analyses on postoperative recovery quality have combined the QoR-15 and QoR-40 scales for comparison (Opioid-free anaesthesia and postoperative quality of recovery: a systematic review and meta-analysis with trial sequential analysis, DOI: 10.1016/j.accpm.2024.101453). This includes studies published in top anesthesiology journals such as BJA (The impact of perioperative ketamine or esketamine on the subjective quality of recovery after surgery: a meta-analysis of randomised controlled trials, DOI: 10.1016/j.bja.2024.03.012). These pieces of evidence support the scientific reliability of combining the two scales for comparison

Reviewer #3: (No Response)

**Do you want your identity to be public for this peer review?** For information about this choice, including consent withdrawal, please see our Privacy Policy

Reviewer #1: No

Reviewer #3: No

---

## [Editor Report · Acceptance letter]

PONE-D-25-33203R1

PLOS ONE

Dear Dr. Wang,

I'm pleased to inform you that your manuscript has been deemed suitable for publication in PLOS ONE. Congratulations! Your manuscript is now being handed over to our production team.

Kind regards,

on behalf of

Dr. Benjamin Benzon

Academic Editor

PLOS ONE